# A Survey of Heart Anomaly Detection Using Ambulatory Electrocardiogram (ECG)

**DOI:** 10.3390/s20051461

**Published:** 2020-03-06

**Authors:** Hongzu Li, Pierre Boulanger

**Affiliations:** Computing Science Department, University of Alberta, Edmonton, AB T6G 2R3, Canada; pierreb@ualberta.ca

**Keywords:** Review, ECG, Signal Processing, Machine Learning, Cardiovascular Disease, Anomaly Detection

## Abstract

Cardiovascular diseases (CVDs) are the number one cause of death globally. An estimated 17.9 million people die from CVDs each year, representing 31% of all global deaths. Most cardiac patients require early detection and treatment. Therefore, many products to monitor patient’s heart conditions have been introduced on the market. Most of these devices can record a patient’s bio-metric signals both in resting and in exercising situations. However, reading the massive amount of raw electrocardiogram (ECG) signals from the sensors is very time-consuming. Automatic anomaly detection for the ECG signals could act as an assistant for doctors to diagnose a cardiac condition. This paper reviews the current state-of-the-art of this technology discusses the pros and cons of the devices and algorithms found in the literature and the possible research directions to develop the next generation of ambulatory monitoring systems.

## 1. Introduction

According to the World Health Organization (WHO), cardiovascular diseases (CVDs) are the number one cause of death globally. An estimated 17.9 million people die from CVD, representing 31% of all global deaths. Four out of five CVD deaths are due to heart attacks and strokes, and one third of these deaths occurs prematurely in people under 70 years of age [1]. An electrocardiogram (ECG) can record a patient’s heart electrical signal activities over a long period [2] by measuring voltages from electrodes attached to the patient’s chest, arms, and legs. ECGs are a quick, safe, and painless way to check for heart rate, heart rhythm, and signs of potential heart disease.

A twelve lead ECG is today’s standard tool and is used by cardiologists for detecting various cardiovascular abnormalities. However, heart problems may not always be observed on a standard 10-second recording from the 12-lead ECG measurements performed in hospitals or clinics. Therefore, long term ECG monitoring that tracks the patient’s heart condition at all times and under any circumstance has become possible with the development of new sensing technologies. Portable ECG recording devices such as the Apple Watch [3], AliveCor [4], Omron HeartScan [5], QardioMD [6], and, more recently, the Astroskin Smart Shirt [7] are revolutionizing cardiac diagnostics by measuring a patient’s 24/7 cardiac activities and transmitting this information to a cloud service to be stored and processed remotely.

By itself, this massive data set is not very useful to the medical community as they usually do not have enough time or resources to read through long ECG recordings (two to three weeks) to detect any possible heart problems. For this technology to work, new automatic and reliable heart anomaly detection algorithms must be developed to assist doctors in coping with this massive data set. Our aim in this paper is to review the current state-of-the-art that address the challenges of performing ambulatory ECG anomaly detection and to highlight possible solutions. In order to do so, we will first review the medical background associated to ECG analysis and then review the state-of-the-art of automated anomaly detection in ambulatory and non-ambulatory contexts. We will then conclude by discussing the possible research directions to develop the next generation of ambulatory monitoring systems.

## 2. ECG Monitoring and Its Signals

### 2.1. Standard 12-lead ECG

A standard 12-lead electrocardiogram provides views of the heart in both the frontal and horizontal planes and views the surfaces of the left ventricle from 12 different angles. A 12-lead ECG has six limb leads (I, II, III, aVF, aVL, and aVR), and six chest leads (V1–V6). The standard 12-lead ECG is used as a standard clinical dysrhythmia analysis tool for chest pain or discomfort, electrical injuries, electrolyte imbalances, medication overdoses, ventricular failure, stroke, syncope, and unstable patients. It is widely used in clinics and hospitals for heart disease diagnosis [8]. However, when the patient needs to be monitored continuously, a 12-lead ECG is impractical as the patient needs to be attached to 10 electrodes.

### 2.2. Three-lead vs. 12-lead ECGs

Due to the fact that the standard 12-lead ECG is impractical for continuous ECG recording, therefore, three (3)-lead ECGs are widely used in portable ECG devices for a 24-h recording. Frank’s lead system [9] is a 3-lead system that is practical for clinical use. In addition, much research has been done to show that a 3-lead ECG is useful to make a valid diagnosis. Antonicelli [10] was able to validate the accuracy of a 3-lead telecardiology (tele)-ECG compared to a 12-lead tele-ECG in an older population. Their study demonstrated a high level of concordance between the ECG diagnosis using a simple home telecardiology device (3-lead tele-ECG) and more complex instruments, like the 12-lead tele-ECG, as well as the standard 12-lead ECG. The study also demonstrated that a simple 3-lead tele-ECG could be used to detect cardiac alterations, such as arrhythmias, atrioventricular blocks, and re-polarization abnormalities, with good agreement with the observations measured by a 12-lead tele-ECG and the standard 12-lead ECG.

Kristensen et al. also evaluated how well an inexpensive portable three-lead ECG monitor (PEM) can detect patients with atrial fibrillation (AF) compared to a standard 12-lead ECG [11]. In their study, the results demonstrated that the sensitivity of diagnosing AF using PEM recordings was 86.7% and the specificity was 98.7% when compared to a 12-lead ECG. According to cardiologists, the misclassification of three PEM recordings was due to interpretation errors and not related to the PEM recording. In their article, they concluded that PEM devices could be used to diagnose AF. Dehnavi et al. performed an analysis of 3-lead vectorcardiogram (VCG) signals for the detection of cardiovascular diseases [12]. In the study, the authors experimented with detecting ischemia using a VCG algorithm using 3-lead and 12-lead ECGs and demonstrated a similar performance in both cases.

Furthermore, many researchers have tried to reconstruct a 12-lead ECG signal from 3-lead signals. Piotr Augustyniak reviewed and compared two transformation functions between 3-lead VCG and 12-lead ECGs [13]: the Dower and Levkov transformations. The author then tested how a synthesized 12-lead ECG and a VCG compared to an actual 12-lead ECG. The results showed that the synthesized 12-lead ECG was 10.08% distorted and the synthesized vectocardiogram was 6.347% distorted. Atoui [14] introduced a neural network-based model that could derive a standard 12-lead ECG from a serial 3-lead ECG. As a result, the derived 12-lead ECG from the ANN model has an average correlation coefficient of 0.93 compared to the actual 12-lead ECG.

Figueriedo et al. proposed using a 3-signal-lead sensor to synthesize a 12-lead ECG [15]. The authors used a linear equation to combine the collected signals from a 3-signal-lead sensor to output the 12-lead ECG. H. Zhu et al. proposed a novel, lightweight synthetic method [16], which could reconstruct the standard 12-lead ECG from 3-leads: I, II, and V2. The proposed method is called the adaptive region segmentation-based piece wise linear (APSPL) method. It consists of adaptive region segmentation, linear regression operation, and ECG sequence restoration.

Moreover, Nelwan et al. [17] and Drew et al. [18,19] have done several studies demonstrating that it is possible to reconstruct a standard 12-lead ECG from a reduced lead set ECG. I. Tomasic et al. performed a study [20] to investigate how a regression trees algorithm can be used to transform a 3-lead ECG into a synthesized 12-lead ECG. Their study demonstrated that the regression trees algorithm can synthesized an accurate 12-lead reconstruction and that the reduced ECG lead set, contains enough information to detect most heart anomalies.

### 2.3. Normal ECG Signals

To detect anomalies on ECG signals, one must first know what a normal heartbeat looks like. In [8], a normal rhythm (see Figure 1) is defined as the result of an electrical impulse that starts from the sinoatrial (SA) node, propagates through the heart muscles, and then to the patient’s chest. A normal rhythm is composed of the following segments in sequence: a P wave generated by the atrial depolarization, the QRS complex generated by the ventricular depolarization, and a T wave and U wave generated by ventricular re-polarization. In normal ECG signals, the P wave, QRS complex, and T wave should be similar over time at a frequency ranging from 60 to 100 bpm. A normal ECG signal should have paced rhythm (PR) intervals within 0.12–0.2 s, and QT intervals less than half of the corresponding RR interval. Also, in a normal ECG signal, the variation between the shortest PP interval/RR interval and the longest PP interval/RR interval should be less than 0.04 s (see Figure 2).

### 2.4. Abnormal ECG Signals

The anomalies in ECG signals can be categorized into three subsets: irregular heart rate, irregular rhythm and ectopic rhythm. The heart rate could be counted by measuring the PP/RR intervals on the ECG. If the PP/RR interval is long, this indicates a low heart rate, otherwise, it indicates a high heart rate. If the heartbeats start from SA node, but the PP/RR intervals are longer than 1 s, this may indicate sinus bradycardia (Figure 3a), which indicates that the heart is pumping too slow. When the PP/RR intervals are shorter than 0.6 s, this may be the sign of Sinus Tachycardia (Figure 3b). Moreover, if the variations between the PP/RR intervals are too large, this may indicate Sinus Arrhythmia, Sinus Block, and Sinus Arrest (Figure 3c–e).

These ECG anomalies may indicate a patient’s current conditions. For instance, Sinus Bradycardia may be associated with hypothyroidism, hyperkalemia, sick sinus syndrome, sleep apnea syndromes, carotid sinus hypersensitivity syndrome, and vasovagal reactions. Sinus Tachycardia is commonly associated with anxiety, excitement, pain, drug reactions, fever, congestive heart failure, pulmonary embolism, acute myocardial infarction, hyperthyroidism, pheochromocytoma, intravascular volume loss, and alcohol intoxication or withdrawal. Sinus Block, and Sinus Arrest can be caused by hypoxemia, myocardial ischemia or infarction, digitalis toxicity, and a toxic response to drugs [22].

Even if the heartbeat starts from the SA node, the heartbeat signal shape could be abnormal as well. For example, in the ECG signal, the ST segment and the T wave could have abnormal shapes; these are usually called ST-T changes. The ST-T changes could indicate hyperkalemia, ischemia, and so on [23]. Some examples of ST-T changes can be found in Figure 4.

Ectopic rhythms are started from a source other than the sinus node. For example, Atrial Rhythms begin in the atria. In this case, the P wave is shaped differently from the P wave beginning in the SA node. There are several abnormal rhythms that can occur when the Atria is firing the heartbeat: Premature Atrial Contraction, Wandering Atrial Pacemaker, Atrial Tachycardia, Atrial Flutter, Atrial Fibrillation. Examples are shown in Figure 5. The Premature Atrial Contraction is a very common heartbeat that could be caused by emotional stress, excessive intake of caffeine, and hyperthyroidism. If Premature Atrial Contraction consecutively occurs three or more times, the rhythm is considered as Atrial Tachycardia. It may cause light-headiness or even fainting. Atrial Flutter and Atrial Fibrillation are two distinct but closely related tachyarrhythmias. They could lead to many symptoms, such as palpitations, light-headiness, fainting, angina, and congestive heart failure.

Junctional Rhythms are another kind of ectopic rhythm. These occur when the atrioventricular (AV) junction paces the heart. In such a case, the P wave on the ECG signal may disappear or become negative. There are several anomaly examples shown in Figure 6: Premature Junctional Complex, Junctional Escape Rhythm, Junctional Tachycardia. The Premature Junctional Complex usually has the same cause as the Premature Atrial Contraction described previously. A Junctional Escape Rhythm could be caused by sick sinus syndrome, digitalis toxicity, excessive effects of beta-blockers or calcium channel blockers, acute myocardial infarction, hypoxemia, and hyperkalemia. One of the most common anomalies is the Junctional Tachycardia, the Atrioventricular nodal re-entrant tachycardia (AVNRT). This is an arrhythmia that results from a rapidly recirculating impulse in the nodal part of the AV junction, and could be caused by digitalis toxicity [22].

Ventricular Rhythms is another kind of ectopic rhythm. It occurs when an ectopic site within a ventricle assumes responsibility for pacing the heart. As a result, the ventricular heartbeats and rhythms usually have QRS complexes that have abnormal shapes and longer lengths. The following are the examples of abnormal Ventricular Rhythm: Premature Ventricular Contraction, Ventricular Escaped Rhythm, Accelerated Idioventricular Rhythm, Ventricular Tachycardia, and Ventricular Fibrillation, Ventricular Asystole. We can see the ECG signals in Figure 7. Individuals with Premature Ventricular Contraction may have the marker of severe organic heart disease associated with an increased risk of cardiac arrest and sudden death from Ventricular Fibrillation. Ventricular Tachycardia consists of three or more consecutive Premature Ventricular Contraction, and it could lead to more life-threatening Ventricular Fibrillation. With Ventricular Fibrillation, the ventricles do not heartbeat in any coordinated fashion but instead, fibrillate or quiver asynchronously and ineffectively. It will cause the patient to become unconscious immediately [22].

As depolarization and re-polarization are slow in the atrioventricular (AV) node, this area is vulnerable to blocks in conduction. Therefore, when a delay or interruption happens during impulse conduction from the atria to the ventricle, AV blocks may occur. AV blocks, also called Heart blocks, are classified into: First-degree AV blocks; Second-degree AV blocks (types I and II); Third-degree AV blocks (complete) see Figure 8. Among the heart blocks, the lower degree heart blocks could lead to Third-degree AV blocks, also called Complete Heart blocks, which are the most severe heart anomaly. With the Complete Heart blocks, the atria and ventricle are pacing independently, which could slow down the ventricular rate, and eventually lead to fainting [22].

## 3. Automatic Heart Anomaly Detection: A State-of-the-Art

### 3.1. Automatic Heart Anomaly Detection

The objective of detecting anomalies in ECG signals consists of finding the irregular heart rates, heartbeats, and rhythms. To achieve this goal, an anomaly detection system must be able to find them on all heartbeat sequences; therefore, to obtain the essential metrics as stated in Section 2. Also, the system looks at the entire recording to detect any irregular rhythm segments such as an inconsistent R-R interval and ectopic rhythms. Therefore, an anomaly detection system is composed of five different sub-systems: noise removal (Section 3.2), heartbeat detection (Section 3.3), heartbeat segmentation (Section 3.3), heartbeat classification (Section 3.4), and rhythm classification (Section 3.5).

A typical heartbeat anomaly detection system can be seen in Figure 9. The noise reduction process intends to minimize its effect on signal interpretation caused by the recording device or patient’s movement. The heartbeat detection aims to find the location of the heartbeats to calculate the heart rate. The heartbeat segmentation extracts the entire heartbeat based on a known heartbeat location. The heartbeat classification checks for any abnormal heartbeat shape on the ECG signal. The irregular heart rhythm classification is similar to the heartbeat classification, but instead of checking only one heartbeat shape, it checks a period signal on the ECG record. Pertinent research found in the literature relating to the five sub-systems are introduced in the following sections.

#### 3.1.1. MIT-BIH Database

Before explaining the sub-systems of the anomaly detection, the MIT-BIH Arrhythmia Database [24,25] need to be described first, as it is widely used in the ECG analysis related research. It was the first generally available set of standard test materials for the evaluation of the arrhythmia detector. It contains 48 half-hour excerpts of two-channel ambulatory ECG recordings from 47 subjects. The ECG data was collected with Del Mar Avionics model 445 two-channel reel-to-reel Holter recorders. The database has the annotation labels for 16 different heartbeat types and 15 different types of rhythms. All the selected research in this review used the MIT-BIH database, which allows us to test and compare the performance of the algorithms.

The annotation labels and the corresponding heartbeat types and rhythm types used in this database are listed below. The heartbeat types are:Normal (N)Left bundle branch block beat (LBBB)Right bundle branch block beat (RBBB)Atrial premature beat (PAC/APC)Aberrated atrial rremature beat (a)Nodal(junctional) premature beat (J)Supraventricular premature beat (S)Premature ventricular contraction (PVC)Fusion of ventricular and normal beat (F)Atrial escape beat (e)Nodal (junctional) escape beat (j)Ventricular escape beat (E)Paced beat (P)Fusion of paced and normal beat (f)Classifiable beat (Q)Atrial/Ventricular flutter beat (!).

The rhythm types are:Atrial bigeminy (AB)Atrial fibrillation (AF)Atrial flutter (AFL)Ventricular bigeminy (B)2${}^{\circ }$ Heart block (BII)Idioventricular rhythm (IVR)Normal sinus rhythm (NSR)Nodal (A-V junction) rhythmPaced rhythm (PR)Pre-excitation (PREX)Sinus bradycardia (SBR)Supraventricular tachyarrhythmia (SVTA)Ventricular trigeminy (T)Ventricular flutter (VFL)Ventricular tachycardia (VT).

## 3.2. Noise Removal

ECG signals may be distorted by many other artifacts that have nothing to do with the heart functions. The ECG artifacts have various and uncertain forms. Some physiologic artifacts could mimic true dysrhythmia, leading to false diagnostics [26]. Therefore, noise removal is a necessary step for anomaly detection in ambulatory ECGs.

There are two main groups of artifacts: non-physiological and physiological artifacts. The first is caused by equipment problems, such as power-line interference, and the other one is caused by muscle activities, skin interference or body motion such as baseline wander, electromyogram, and motion artifacts. For example, the motion wander could significantly affect the measurement of the ST segment in an ECG signal [27]. Among all the artifacts, the motion artifact is the most challenging noise to remove as the noise spectrum overlaps the ECG signal [28]. Various ECG motion artifact examples are shown in Figure 10 and Figure 11 [29].

In this section, various noise removal algorithms in the research literature are categorized and compared. There are four conventional methods used for noise removal in ECG signals.

The first approach consists of using digital low-pass, high-pass, band-pass, and notch filters to remove the noise. Many studies, such as [30,31,32,33,34,35], use a combination of low-pass and high-pass filters to remove the corresponding noise on an ECG signal. The low-pass filter cut-off frequency is in the range of 11 Hz to 45 Hz, and it mainly suppresses the high-frequency noise. The high-pass filter cut-off frequency is in the range of 1 Hz to 2.2 Hz, and it focuses on removing the baseline wander in the signal. In [30,32], notch filters range from 50 Hz to 60 Hz and are used for removing the power-line interference. Band-pass filters with cut-off frequencies from 0.1 to 100 Hz are used by [36] to remove the noisy components of electronic noise. The advantage of using a fixed digital filter is that it is easy to implement and is highly efficient.

The second approach is to use a discrete wavelet transform (DWT) to remove the noise components from a signal. Wavelet transform is a powerful method for analyzing non-stationary signals, such as ECGs [37]. The DWT noise removal method is used in [38,39,40]. This method decomposes the signal into the approximation and detail coefficients by using a wavelet function. The selection of the wavelet function in the wavelet transform is the most important task, which depends upon the type of signal [41]. The commonly used Mother Wavelet basis functions are Daubechies filters (Db), Symmlet filters (Sym), Coiflet filters (C), Battle-Lemarie filters (Bt), Beylkin filters (Bl), and Vaidyanathan filters (Vd) [42].

According to studies in [41,42,43], the Daubechies filters of order 4 and 8 (Figure 12), and the Symmlet filters of order 5 and 6 (Figure 13) are the best wavelet functions for ECG signal analysis due to their similar signal structure to the QRS complex. After decomposing the ECG signal, a threshold method is applied to the DWT coefficients. A clean ECG signal could be reconstructed from the thresholded DWT coefficients.

DWT relies on the choice of the wavelet basis [44]. The level of DWT may be different between different data sets; therefore, re-implementation is needed. Another wavelet analysis method is the empirical mode decomposition (EMD). The EMD is an adaptive and fully data-driven technique that obtains the oscillatory modes present in the data [44]. The EMD, similar to the wavelet analysis, decomposes a time series signal into individual components without leaving the time domain. In EMD, the high-frequency components are called the intrinsic mode function (IMF), and the low-frequency part is called the residual. The procedure can be applied to residuals iteratively until no IMFs can be extracted. The IMFs must satisfy two conditions:The number of extremas and zero-crossings must be equal or differ at most by one;All local maximas and minimas must be symmetric to zero.

After decomposition using EMD, an IMFs and one residual signal will be obtained. Let c(t) be the IMFs, we will have $c_{1}\left(t\right)$ to $c_{n}\left(t\right)$ from higher frequency components to lower frequency components. Then digital filters or thresholds can be applied to the IMFs that contain the noise. After processing, the signal can be reconstructed using the following equation:(1)$$x\left(t\right)=\sum_{i=1}^{n}c_{i}\left(t\right)+r\left(t\right)$$
where x(t) is the reconstructed signal, c(t) is the IMFs, and r(t) is the residual signal.

In [45,46,47], the authors performed an EMD on the MIT-BIH database to suppress the high frequency noise and the baseline wander. Ensemble empirical mode decomposition (EEMD) [48] fixed the EMD shortcoming of mode mixing. The mode mixing can cause serious aliasing in the time-frequency distribution, and also makes the physical meaning of individual IMF unclear. The EEMD adds one extra step comparing to the EMD. By adding white noise to the original signal before decomposing the signal into IMFs using EMD. Many noise removal works were found using the EEMD, such as [49,50,51].

The previous approaches work well when the noise is in a fixed frequency range. However, there are some cases where these approaches could fail. The first one is in motion wander removal. Raimon Jane et al. stated in [27] that the motion wander frequency may not always be below 0.05 Hz. It could depend on the frequency of the heart rate, which could be less than 0.8 Hz. Also, a fixed digital filter could introduce nonlinear phase distortion and key point displacement [52]. These two approaches could not remove the motion artifact from the ECG signal, as its spectrum completely overlaps with the ECG signal. Therefore, many approaches use adaptive filtering to solve the proposed problem.

In 1991, Thakor et al. [28] introduced the least mean squares (LMS) adaptive filter (ARF) to reduce the baseline wander, 60 Hz power line noise, muscle noise, and motion wander. In their research, two adaptive filter structures were proposed. The first one has the primary input as $s_{1}+n_{1}$, while the reference input is noise, $n_{2}$, which could be recorded from another generator that is correlated with $n_{1}$. The second one is an ECG that is recorded from several electrode leads, the primary input is $s_{1}+n_{1}$ from one of the leads, the reference input is $S_{2}$ from another lead that is noise-free. In both cases, the signal $s_{1}$ can be extracted by recursively minimizing the mean squared error (MSE) between the primary and the reference inputs. The MSE can be calculated as:(2)$$E\left[\epsilon^{2}\right]=E\left[{\left(s_{1}-y\right)}^{2}\right]+E\left[N_{1}^{2}\right].$$
The least mean squares (LMS) algorithm was used to minimize the MSE. The LMS algorithm could be written as:(3)$$W_{k+1}=W_{k}+2\mu \epsilon_{k}X_{k}$$
where $W_{k}$ is a set of filter weights at time *k*, $X_{k}$ is the input vector at time *k* of the samples from the reference signal, ϵ= primary input $d_{k}-$ filter output *y*, and parameter μ is empirically selected to produce convergence at a desired rate. The error $\epsilon_{k}$ can be calculated as:(4)$$\epsilon_{k}=d_{k}-y_{k}$$
where $d_{k}$ is the desired primary input from the ECG to be filtered, and $y_{k}$ is the filter output that is the best least squares estimate of $d_{k}$.

As LMS adaptive filters are sensitive to scaling of the input, a power normalized least mean squares has been introduced to solve this problem [53]. Another convention adaptive filter type is the recursive least square (RLS) adaptive filter. The RLS algorithm recursively finds the filter coefficients that minimize a weighted linear least-squares cost function relating to the input signal. It is known for its excellent performance when working in time-varying environments but at the cost of increased computational complexity and some stability problems [54]. The algorithm updates the filter weight vector using the following equations:(5)$$w\left(n\right)={\bar{w}}^{T}\left(n-1\right)+k\left(n\right){\bar{e}}_{n-1}\left(n\right),$$
(6)$$k\left(n\right)=u\left(n\right)/(\lambda +x^{T}\left(n\right)u\left(n\right)),$$
(7)$$u\left(n\right)=\bar{w_{\lambda }^{-1}}\left(n-1\right)x\left(n\right),$$
where w(n) is the weights vector of iteration n, x(n) is the input signal, and λ is a small positive constant very close to but smaller than 1.

The filter output ${\bar{y}}_{n-1}\left(n\right)$ and the error signal ${\bar{e}}_{n-1}$ are calculated using the filter tap weights of the previous iteration and the current input vector as in the following equations:(8)$${\bar{y}}_{n-1}\left(n\right)={\bar{w}}^{T}\left(n-1\right)x\left(n\right),$$
(9)$${\bar{e}}_{n-1}=d\left(n\right)-{\bar{y}}_{n-1}\left(n\right).$$
An adaptive filtering approach could remove baseline wander, motion artifacts, power-line interference, and the muscle noise; however, it requires a reference input that is correlated to the original noisy input. Obtaining a clean ECG signal is very difficult to acquire. Due to the added complexity for the data collection, many studies have considered using an accelerometer as the reference noise signal for the adaptive filter. For example, in [55], Raya et al. explored the possibility of using both a signal axis and dual-axis accelerometer signal as the noise reference input to a least mean square (LMS) adaptive filter and a recursive least square (RLS) adaptive filter. As a result, the RLS adaptive filter outperformed the LMS adaptive filter. Using an accelerometer signal showed better results than using a dual-axis accelerometer signal. The authors believed that the use of one axis reference input, particularly the y-axis, was sufficient to minimize the noise.

## 3.3. Heartbeat Detection and Segmentation

Heartbeat detection is often related to the detection of an irregular heart rate and inconsistent RR-intervals, which are explained in Section 2. Heartbeat detection is also the key step to extract the heartbeats from the ECG signal to be used for classification. Heartbeat detection consists of three main parts: P wave detection, QRS complex detection, and T wave detection. Therefore, it is usually related to heartbeat segmentation. Heartbeat segmentation usually means segmenting a heartbeat from its start point (onsite) of P wave to its endpoint (offsite) of the T wave.

However, the P wave and T wave may not be detectable in certain types of abnormal heartbeat, and the QRS complex is the most obvious waveform. Thus the location of the QRS complex is often used to locate the origin of the heartbeat; see Figure 2. There are many studies that detect the R peak location in the QRS complex.

The Pan–Tompkins algorithm [56] is one of the most popular and earliest algorithms that has been implemented (Figure 14). It is widely used in many applications due to its robustness and computational efficiency. The algorithm uses a filter bank that consists of band-pass filters, a differentiator, a squaring filter, and a moving window integrator to reduce the signal noise so that only R wave information is present. Inspired by the Pan–Tompkins algorithm, many researchers, such as [57,58,59,60] developed their own filter banks to improve the accuracy of the detection. In order to reduce the detection of false positives, [58,60] used a predefined amplitude threshold, [59,60] used a predefined RR interval length threshold.

Zidelmal et al. introduced a QRS detection method based on wavelet decomposition [61]. In the algorithm, the authors decomposed the raw ECG signal using a discrete wavelet transform, then reconstructed the signal by selecting only the sub-signals that contained ECG information. To detect the QRS complex, a threshold was set to select the peaks that have a large amplitude. Similar works have been done in [62].

Manikandan et al. introduced a new algorithm that uses a Shannon energy envelop and Hilbert-transform (SEEHT, Figure 15) to detect the QRS complex location [63]. In the preprocessing stage of their algorithm, a band-pass filter is applied to the raw ECG signal to remove the baseline wander and high-frequency noise. After that, a differentiator and normalizer is applied to the clean signal to highlight the QRS complex components. The Shannon energy of the processed signal is calculated using the following equation:(10)$$s\left[n\right]=-d^{2}\left[n\right]log\left(d^{2}\left[n\right]\right),$$
where d[n] is the processed signal. The calculated Shannon energy sequence is then processed by a zero-phase filter to preserve the sharp peaks around the QRS complex and smooth out the noisy peaks. In the peak finding algorithm, a Hilbert transform is applied on all the candidate R peaks to obtain the R wave envelope. In each R wave envelope, the zero-crossing locations indicate an R peak.

Inspired by SEEHT, [64] introduced An R-peak detection method based on peaks of Shannon energy envelope(PSEE) that improves the computational inefficiency of the Hilbert transform by using both predefined amplitude thresholds and predefined RR interval length thresholds. An improved R-peak detection method based on Shannon energy envelope (ISEE) [65] improved further the SEEHT and PSEE algorithms by using a filter bank consisting of a moving average filter, a differentiator, a normalizer, and a squaring filter to eliminate the noisy peaks. The filter bank computational costs is less than the Hilbert transform and does not use a predefined threshold. Most recently, Park combined discrete wavelet transform and ISEE to detect R peaks on the ECG signals [66].

As explained previously, the P and T waves represent important information and the heartbeat segmentation depends on the P and T wave detection. Therefore, a good detection of the P and T waves is critical for diagnosis. Pal and Mitra proposed an algorithm that could detect the PQRST peak points [67]. The algorithm is based on discrete wavelet decomposition. It reconstructs the signal from selected wavelet coefficients, which are related to peaks such as: R, QS, and PT. For example, when the algorithm is detecting the R peak, a signal is reconstructed with d3, d4, and d5 coefficients, and this preserves the information for the R peaks but diminishes the other peaks.

A few years later, Banerjee also developed a T wave and QRS complex detection algorithm based on discrete wavelet decomposition and adaptive thresholding [68]. Karimipour uses discrete wavelet transform and adaptive thresholding to detect the QRS complex location, and give an estimate of the P-wave and T-wave locations [69]. In practice, many studies, such as [31,70] used the ’ecgpuwave’ detector from PhysioNet for heartbeat segmentation [25]. However, because the P and T wave detection works well with normal heartbeats, but not for many abnormal heartbeat types. Many researchers choose manual annotation, such as [71], or a fixed window, such as [32,33,38,71,72], for their heartbeat segmentation.

In Table 1, we compare the performance of some of the heartbeat detection algorithms that have been tested on the MIT-BIH Arrhythmia database. [24].

The metrics used to compare each algorithm are calculated as follow:TP: Number of correctly detected heartbeatsFP: Number of incorrectly detected heartbeatsFN: Number of missed heartbeatsSensitivity (SEN) = TP / (TP+FN)Positive Detection (+P) = TP / (TP+FP)Detection Error Rate(DER) = (FP+FN) / TPAccuracy (ACC) = TP / (TP+FP+FN).

## 3.4. Irregular Heartbeat Classification

Irregular heartbeat classification focuses on the shape of the heartbeats, and aims at classifying the type of a single heartbeat. As discussed previously, the heartbeat shape may vary when the heartbeat starts from an ectopic location. For example, a premature heartbeat may have a missing P wave. The abnormal shape of a heartbeat may indicate potential heart disease. By classifying and annotating the types for all the heartbeats on the ECG, one could easily notice the frequency of anomalies that happens in the heart to make an appropriate diagnosis and treatment. Heartbeat classification consists of two main parts: feature extraction and model training.

### 3.4.1. Feature Extraction

The feature extraction step converts the raw ECG signal to machine-readable information. Based on the existing research, there are two common features: morphological features and derived features. The morphological features describe the heartbeats based on the observations of the signal itself. There are many morphological features (see in Table 2) that have been used in various studies.

Other derived features are calculated from the ECG signal. There are many different methods are proposed in the literature:Vectorcardiography (VCG) vector;DWT coefficients produced by Discrete Wavelet Transform (DWT);Independent components from Independent Component Analysis (ICA);PCA components generated from Principal Component Analysis (PCA);IMFs from Empirical Mode Decomposition (EMD)/Ensemble EMD (EEMD);DTCWT coefficients from Dual Tree Complex Wavelet Transform;Eigenvector methods;Dynamic Time Warping (DTW) distance.

Vectorcardiography (VCG) is one of the ECG analysis tools. It displays the various complexes of the ECG. It provides the possibility to use vector analysis on the cardiac electric potentials [80].

Discrete Wavelet Transform (DWT) decomposes the signal into many sub-signals (detail coefficients) with different frequency ranges, as described in Section 3.2. Not only could the DWT method be used to remove unwanted noises, it could also find features for the heartbeats as the heartbeat waves are much clearer in the specific detail coefficients, such as D4 and D5. Therefore, much research, such as [38,71], uses features from the detail coefficients to classify the heartbeat.

The conventional DWT technique lacks the property of shift-invariance due to the downsampling operations at each stage of DWT implementation. Hence, the energy of the wavelet coefficient changes significantly for a small-time shift in the input pattern. The Dual-Tree Complex Wavelet Transform [81] is a simple technique that overcomes the DWT shortcomings. The DTCWT uses two sets of filters: one is used for level 1 decomposition, and the other one is used for the higher levels. In the first level decomposition, the original signal is decomposed into two Trees, and each Tree contains two sub-band signals. One tree could be interpreted as the real part of a complex wavelet, and the other tree could be the imaginary part. For each tree, the conventional DWT is applied for further decomposition [32]. The DTCWT method was used by Thomas to extract heartbeat features to classify the heartbeat type [32].

Similar to DWT and DWCWT, the ICA, PCA, and EMD/EEMD also decompose the signal into many sub-signals. The difference is that the ICA and PCA aims to reduce the input size to minimize the computation speed. The EMD/EEMD, as explained in Section 3.2, does not require the knowledge of the level of scale and the basis function that is needed in DWT. The ICA method has been used in [71] to produce the independent components to be part of the heartbeat feature set. The PCA method used in [82] reduces the input size for higher efficiency. Rajesh et al. computed the heartbeat features from IMFs by applying the EMD/EEMD method to the ECG signal.

Eigenvector methods are used for estimating the frequencies and powers of signals from noise-corrupted measurements. These methods are based on an eigendecomposition of the correlation matrix of the noise-corrupted signal [83]. In [83], Ubeyli et al. used three kinds of eigenvector methods to generate the feature set: Pisarenko, Multiple Signal Classification (MUSIC), and Minimum-Norm. The Pisarenko method is particularly useful for estimating a PSD that contains sharp peaks at the expected frequencies. The MUSIC method is a noise subspace frequency estimator and could eliminate the effects of spurious zero on the noise subspace. The Minimum-Norm method aims to differentiate spurious zeros from real zeros, and it uses a linear combination of all noise subspace eigenvectors.

Dynamic Time Warping measures the similarity between two heartbeat segments. It computes the distance between these two heartbeat segments. Therefore, if we let one of the heartbeat segments to be the sample heartbeat of a specific type, and the other one to be the test heartbeat, then the distance indicates the similarity score between the test heartbeat and the sample heartbeat. The similarity score could be used as a feature that represents the heartbeat, such as in work by [74,76]. Details of the features of each method reviewed can be seen in Table 3.

### 3.4.2. Model Training

Once the feature vectors are extracted from the raw ECG signal, then they can be used by a model for training and classification. There are several methods that have been proven to be valid for identifying heartbeat types. They are clustering, traditional machine learning classification, and deep learning classification.

The clustering aims to find the similarity between the two groups (heartbeat segments) by computing the distance between the two groups. The conventional distances for ECG signals are the Euclidean Distance and Dynamic Time Warping Distance. [84].

The Euclidean distance is the most common distance when comparing two groups with the same dimensions. An example of using Euclidean distance for abnormal heartbeat detection can be found in Chuah and Fu’s [76]. They introduce an adaptive window discord discovery (AWDD) to detect the anomaly in ECG recordings. It was developed from a brute force discord discovery (BFDD) algorithm [85]. The algorithm finds candidates with an abnormal heartbeat by selecting the largest Euclidean distance when comparing the heartbeats to each other. Also, they have set a threshold for the Euclidean distance to reduce the false alarm rate. The Euclidean distance only works when both heartbeat segments are the same length.

K-mean clustering is a popular clustering method that builds on the Euclidean distance. The K-mean clustering clusters the heartbeat segments into many different clusters. Veeravalli et al. developed an algorithm for real-time and personalized anomaly detection from wearable health care ECG devices [86]. The K-means cluster algorithm is used to cluster all the heartbeat classes. To avoid calibration of the technique for individual users, they assigned the most frequent heartbeat segments as the normal heartbeat segments. The authors tested their algorithm on the MIT-BIH database and the European ST-T Database. They were able to achieve 97.1% sensitivity and 99.5% specificity.

Sivarake and Ratanamahatana proposed a robust and accurate anomaly detection algorithm (RAAD) that reduced the false alarm detection rate on ECG anomaly detection [34]. They extracted heartbeat morphological features to be their input feature vectors. Then, they calculated the dynamic time warping distance to measure the similarity between two variable-length heartbeats. In their experiment, they tested their algorithm on INCARTDB01-05 [25], the MIT-BIH arrhythmia database [24,25], and the MIT-BIH long term database [25]. Overall, their algorithm achieved 94.35% accuracy and a 0% false alarm rate.

Another major method is the traditional machine learning classification algorithms: Kth nearest neighbor(KNN), Linear Discriminant Analysis(LDA), Quadratic Discriminant Functions(QDF), Support Vector Machine(SVM), and Multilayer perceptron neural network(MLPNN). These algorithms build a mathematical model based on the provided training data. The trained model could correlate the input data with its corresponding label. Many research could be found in this field.

Ivaylo Christov et al. used both the ECG morphology features and VCG features to represent the heartbeat, and then train the feature vectors and its labels with Kth nearest neighbor. As a result, the classification performance on both feature sets is over 96% for five heartbeat types (N, PVC, LBBB, RBBB, and P) [30].

Philip de Chazal et al. used linear discriminant analysis as a classification algorithm. The input feature vectors are ECG morphology features. As a result, this algorithm could perform around 97% accuracy on MIT-BIH database with five heartbeat types (N, S, V, F, and Q) classification [31].

Mariano Llamedo et al. validated a heartbeat classification method for Normal, Supra-ventricular, and Ventricular heartbeats based on ECG interval features, morphological features, and DWT features [38]. The feature vectors are trained with quadratic discriminant functions. The model had a 94% overall classification accuracy on the test dataset.

Li et al. uses the concept of transductive transfer learning to detect the abnormal instance on an ECG signal. They trained a model to learn from a labeled data set to detect irregular heartbeats, and then they use a kernel mean matching (KMM) algorithm [87] to enable knowledge transferring between a labeled data set and unlabeled data set. The model they used was a weighted transductive one-class support vector machine, which could solve the problem of imbalanced data set [78]. The authors performed experiments on records 100, 101, 103, 105, 109, 115, 121, 210, 215, and 232 from the MIT-BIH database. They achieved a 87.89% average accuracy.

Ye et al. classified 16 heartbeat types by using both morphological and dynamic features of ECG signals. Then, both morphological and dynamic features were trained by the support vector machine for the classification. Also, two channels of the ECG signal in the database were trained separately and generated two models. Both models were used for the final classification part. The authors introduced two ways of making a final decision: one is rejection, which requires both models to make the same decision, and the other one is Bayesian, which is based on the fusion of both model’s results [71]. The experiment result of this research is compared in Table 4.

Zhang et al. built 46 feature vectors to represent the heartbeat to classify the abnormal heartbeat shape on MIT-BIH database [74]. In the study, the authors apply the ecgpuwave tool from PhysioNet [25] to detect the boundaries of the P wave, QRS complex, and ST waves. Then they have collected five types of features, which are five inter-heartbeat intervals, five intra-heartbeat intervals, 29 morphological amplitudes, six morphological areas, and morphological distance. The five types of features could generate a feature vector with 46 morphological features. In the classification step, the author used the support vector machine to learn the patterns of the feature vectors. Additionally, both channels of the ECG signal have a trained support vector machine model. The results of both models are considered in the final classification result. The result table of the paper shows that the algorithm has nearly 90% accuracy for four heartbeat types (N, F, V, and S) classification.

Thomas et al. introduced an automatic ECG arrhythmia classification idea using dual-tree complex wavelet-based features to detect normal, paced, RBBB, LBBB, and PVC heartbeats. The authors proposed a feature extraction technique based on a dual-tree complex wavelet transform (DTCWT) technique. Then the feature vectors were input to a multilayer perceptron neural network for abnormal heartbeat detection [32]. The experimental results of this research are compared in Table 4.

Kandala Rajesh et al. used ensemble empirical mode decomposition (EEMD) features to classify normal PVC, PAC, LBBB, and RBBB heartbeats. For the classification tool, a sequential minimal optimization SVM was used to train and classify the different heartbeat types [33]. The experimental results of this research are compared in Table 4.

Wess et al. implemented a multi-layer perceptron (MLP) classifier to detect anomalies in ECG signal. To reduce the size of feature vectors, the author applied PCA on the extracted heartbeats. Finally, the processed feature vectors were used as inputs to train an MLP neural network. The trained model could be used for classifying the anomalies in the ECG signal [82]. The authors were able to test their model on the MIT-BIH database with an overall accuracy of 99.82%.

Most researchers have used traditional methods to solve the problem. Traditional machine learning classification methods do not require a considerable amount of training data, and they do not need a lot of computational power. Recently, due to the development of GPUs, deep learning has been proven to be reliable and fast for classification problems. Compared to traditional algorithms, deep learning does not require cardiology experts to extract features since the network can extract the features automatically. Instead, a deep learning model needs many labeled data for training. Luckily, public data sets could be easily found on the Internet. Therefore, many studies using various deep learning architecture have published new algorithms to classify heartbeats.

The Ubeyli algorithm uses Eigenvectors as the feature vectors and a recurrent neural network as the classification tool. In the experiment, normal, congestive heart failure, VT, and AFIB rhythms were trained and tested [83]. The experiment result of this research is compared in Table 4.

Chauhan and Vig developed a predictive algorithm that could detect normal, PVC, PAC, paced heartbeats via deep LSTM (long short-term memory) neural network (Figure 16). In their algorithm, the features extraction/selection step is neglected, raw ECG data, and corresponding labels are used as inputs to the stacked (two-layer) LSTM neural network. In the experiment, they split the MIT-BIH database into four sets: a non-anomalous training set ($S_{N}$), non-anomalous validation set ($V_{N}$), mixture of both abnormal and normal validation sets ($S_{N+A}$) and the test sets ($t_{N+A}$). The LSTM network was trained on $S_{N}$, and used $V_{N}$ for early stopping. The trained LSTM network was then applied to $S_{N+A}$ to find the threshold for detecting abnormal heartbeats. Finally, the chosen threshold was used on $t_{N+A}$ to discriminate regular and anomalous heartbeats while predicting [79]. The presented model was able to achieve a 97.5% precision with a 46.47% recall on the test set ($t_{N+A}$).

Kiranyaz et al. presented a fast and accurate patient-specific ECG classification and monitoring system. In their experiment setup, they picked five heartbeat types, N, V, S, F, and Q, from 20 ECG records (100–124) from the MIT-BIH database as the training samples. The raw heartbeat segments were submitted to a 1-D adaptive convolutional neural network (CNN) for pattern recognition. The 1-D convolutional neural network acted as a feature extraction tool as well as a classification tool. The classification times for this model is 0.58 and 0.74 ms for 64 and 128 sample heartbeat resolutions, respectively. The speed is more than 1000x faster than the real-time requirement [36]. The experiment result of this research is compared in Table 4.

Sahoo et al. made an improvement to Rai’s algorithm [39] by using multi-resolution wavelet transform and machine learning to detect Normal, LBBB, RBBB, and Paced heartbeats [75]. The authors used Q-peak, R-peak, S-peak, T-peak, QR-interval, ST-interval, RR-interval, and QRS duration as the input feature vector and used a MLP and a SVM classifier as the classification tool. In their experimental results, the overall classification accuracy of normal, LBBB, RBBB, and Paced heartbeats were 96.67% for the SVM classifier and 98.39% for the MLP classifier. The algorithm was tested on the MIT-BIH database [24].

In addition to training with a public data set, some researchers used a patient-specific approach to train the model. The first step of a patient-specific approach is to train an initial classifier with the public data set. Then the second step requires a local cardiologist to review and correct the produced labels by the initial classifier. The final step consists of training the initial classifier with corrected labels to produce the final classifier to this specific patient. The patient-specific approach could eliminate the inter-patient variations of the ECG signals. Biel et al.’s research shows that the variance in different human heartbeats can be very high [88]. Many research works, [31,89,90,91,92,93], have proven that by using a patient-specific model, the detection algorithms have a higher accuracy than the traditional systems in practical cases.

## 3.5. Irregular Rhythm Classification

Different irregular heartbeat classifications can be found in the literature. Rhythm classification focuses on finding abnormal rhythm among normal rhythms. To find a rhythm anomaly, the algorithm needs to process more than one heartbeat.

Ge et al. [94] used an auto-regressive (AR) modeling technique to classify the Normal, PAC, PVC, SVT, VT, and VF rhythms. The algorithm uses Burg’s algorithm to compute the AR coefficients X. In their paper, the authors have attempted two ways to classify the AR coefficients of X: a generalized linear model (GLM) and multi-layer feed-forward neural network. The GLM equation is:(11)Y=Xβ+ε,
where $Y=[y_{1},y_{2},...,y_{N}]$ is an N-dimensional vector of the observed responses, X is the N * P matrix of the AR coefficients, β is a P-dimensional vector, ε is an N-dimensional error vector. The GLM outputs, $y_{1}$ to $y_{N}$, compared to predefined conditions to classify various heartbeat types. An artificial neural network with the AR coefficients as inputs was used for training and classification. Their experimental results show that artificial neural networks perform better than GLM.

Ozbay et al. integrated a type-2 fuzzy clustering and discrete wavelet transform in order to build a neural network-based ECG classifier to detect Normal, Br, VT, SA, PAC, P, RBBB, LBBB, AF, and AFI results [95]. The proposed diagnostic algorithm can distinguish 10 different rhythm types. The system was formed by combining fuzzy clustering layers, feature extraction layers, and a final classifier layer. The fuzzy clustering layer select segments represents the arrhythmia class in the ECG. A wavelet transformation was applied to the ECG segments to generate features. The authors have trained three Type-2 Fuzzy Clustering Neural Network models (T2FCWNN-1, T2FCWNN-2, and T2FCWNN-3) with three different training data sets. The three training data sets have the same amount of ECG segments. However, the length of each ECG segment is 101 sample points, 52 sample points, and 27 sample points. As a result, the T2FCWNN-3 had the lowest training time, which is 4.86 s and test error rate, which is 0.23% among all three models.

Patel et al. used a thresholding technique to detect arrhythmias on ECGs collected from a mobile platform [35]. In the paper, they first used the Pan–Tompkins [56] algorithm to detect the R peaks on the ECG recordings. Then they characterized SB, ST, PVC, PAC, and Sleep Apnea using a predefined threshold to classify different rhythms. Their system a 97.3% detection accuracy.

Rajpurkar et al. developed an algorithm that could out-perform a board-certified cardiologist in the detection of 12 types of arrhythmia using a 34-layer CNN [96]. The network took a 30 s long raw ECG signal recording as input, and the output was a sequence of label prediction. The model output a new prediction every second. The training data set contained 64,121 ECG records from 29,163 patients, and the testing data set contained 336 records from 328 patients. The model performed with 80.9% precision, 82.7% sensitivity, and a 0.809 F1 score.

Acharya et al. used two 11-layer CNNs to detect AFIB, AFL, ad VF(VFL) from normal heartbeat rhythms [40]. The two networks, Net A and Net B, used a 2-second raw ECG recording and 5-second raw ECG recording as inputs, respectively and output the corresponding label. In the algorithm, no wave detection was performed on the input data. Before submission to the 1-D deep CNN, the ECG segments were Z-score normalized. The result of Net A and Net B are compared in Table 5.

### 3.6. Heartbeat/Rhythm Classification Algorithm Comparison

In the previous sections, we reviewed many algorithms that classify the ECGs in various categories. We can see in Table 4 the classification results performed using the MIT-BIH database. In addition, some algorithms’ performance metrics were converted to binary classification, which detects normal and abnormal heartbeats. The reason is that computer diagnoses are not 100% accurate. We still need doctors to make the final diagnosis as they are the only ones who know the context. The methods should be focusing on binary classification, which classifies all abnormal heartbeat as one class. The terms used in the table are explained:TP: Number of correctly detected abnormal heartbeatsFP: Number of incorrectly detected abnormal heartbeatsTN: Number of correctly detected normal heartbeatsFN: Number of incorrectly detected normal heartbeatsSensitivity = TP/(TP+FN)False Alarm Rate= 1 - Specificity = FP/(FP+TN)Accuracy = (TP+TN)/(TP+FP+TN+FN)

Similarly, Table 5 compares all methods that classify the rhythms on MIT-BIH database. In addition, the table has only shown the algorithms that provided enough information to compute our metrics.

## 4. Discussion

### 4.1. Challenges for Heart Anomaly Detection with Ambulatory Electrocardiograms

There are still several challenges in heart anomaly detection:ECG signals may be contaminated with motion noise as the patient is constantly moving. The noisy signal may have a similar morphology to abnormal cardiac signals resulting in false positive. It is easy for the human eye to identify these conditions; however, for computers, it is much harder to separate the noise from the signal.The model training requires a labeled ECG signal. In order to label the ECG data set, trained personnel are needed. In addition, the labeling process is very time consuming. For example, a 10 s one ECG signal has 2500 data points, and the continuous monitoring usually takes 24–48 h.The ECG heartbeat data is highly imbalanced. Over 99% of the heartbeat data is the normal case and only 1% of the heartbeat data presents 16 abnormal cases. Therefore, the highly imbalanced dataset makes it more difficult to adjust the learning step. Several options could be explored to reduce the effect of imbalanced data, such as database re-sampling or using the cost-sensitive method, kernel based method, or active learning [97].

### 4.2. Future Works

The next generation of heart anomaly detection algorithms should be able to deal with ambulatory health measurements taking advantage of multiple synchronized measurements from: accelerometer, real-time blood pressure (based on pulse transit time), skin temperature, and upper and lower chest breathing sensors. An excellent review of the state-of-the-art of body sensor fusion work can be found in Gravina et al. [98]. One commercial example of such a data fusion system is Astroskin from Carre Technologies Inc. The Astroskin space-grade garments offer state-of-the-art continuous real-time monitoring for 48 h of blood pressure, pulse oximetry, 3-lead ECG, respiration, skin temperature, and activity. Using Astroskin, one can develop new fusion algorithms that can compensate for ECG motion artifacts by correlations with synchronized accelerometer and breathing data.

This can be accomplished by using advanced LSTM and recurrent neural networks (RNN). An example of this approach can be found in Shrimanti et al. [99] where ECG, peak blood oxygenation signal (PPG), and accelerometer measurements were combined using a LSTM and RNN to compute in real-time motion compensated blood pressure. Such technology could open the door to real-time patient-specific anomaly detection that goes far beyond simple ECG measurements, for example correlating cardiac and respiratory events with patient activities.

### 4.3. Conclusions

In this survey, we have first introduced the definition of anomaly detection on ambulatory electrocardiograms (ECG) and its importance. We then discussed the basic medical background (Section 2) of electrocardiogram interpretation and the type of anomalies that need to be detected. Most electrocardiogram anomalies can be categorized into two major categories: irregular heart rates and irregular heart rhythms. The irregular heart rates on ECG could indicate bradycardia, tachycardia, heart block, arrhythmia, and so on. The irregular heart rhythms could be ectopic heartbeat when checking a period of ECG signal.

Therefore, based on the different irregularities on the ECG, anomaly detection can be divided into several categories: heartbeat detection (Section 3.3) for detecting the location of each heartbeat; heartbeat segmentation (Section 3.3) for segmenting the heartbeats from the entire ECG signal; heartbeat classification (Section 3.4) for classifying the type of one heartbeat; and rhythm classification (Section 3.5) for classifying the type of a period of ECG signal. In addition, as the ECG signal is frequently contaminated with electrical noise and motion artifacts, noise removal (Section 3.2) is important for anomaly detection on the ambulatory ECG.

From the literature, we have reviewed the conventional methods for each part. For the noise removal on ECG, fixed digital filters, discrete wavelet transform, empirical mode decomposition, and adaptive filters have been used by many researchers. For heartbeat detection, many researchers used fixed digital filters, discrete wavelet transform, and Shannon energy envelopes to remove the noise and unwanted waves while preserving the R peak information. They then used the R peak location to compute the heartbeat. For heartbeat segmentation, the most common method was to use a predefined window to segment the heartbeat signal from the entire signal.

For the literature on heartbeat classification, authors used morphological features and derived features to represent the heartbeat signal. The morphological features were calculated from the ECG signal, and derived features were computed using other methods, such as discrete wavelet transform, independent component analysis, empirical model decomposition, and many more. Both morphological and derived features are then used for training in order to generate a mathematical model of the heartbeat signal.

The most popular models used k nearest neighbor, linear discriminant analysis, support vector machines, multilayer perceptron neural networks, and deep neural networks, such as CNN and RNN. Similarly for the rhythm classification, the algorithms take a period of ECG signal as the input to the model.

The current challenges of anomaly detection for ambulatory electrocardiograms are analyzed in this paper. We determined three major challenges. First, the reduction of motion artifacts on the ECG signal interferes with the anomaly detection. Second, model training requires a massive amount of labeled data that are had to come by. Third, ECG databases have very imbalanced data making it difficult for deep learning model training.

## Figures and Tables

**Figure 1 sensors-20-01461-f001:**
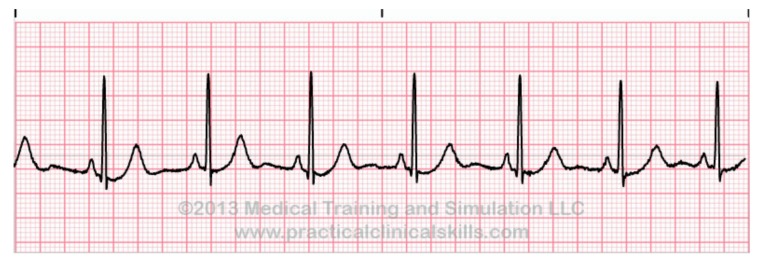
Normal sinus rhythm (NSR) [21].

**Figure 2 sensors-20-01461-f002:**
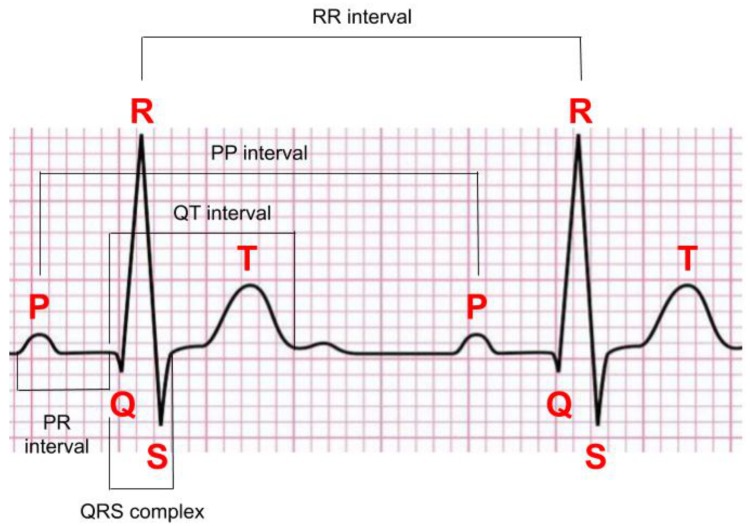
A normal electrocardiogram (ECG) signal and the corresponding notation [21].

**Figure 3 sensors-20-01461-f003:**
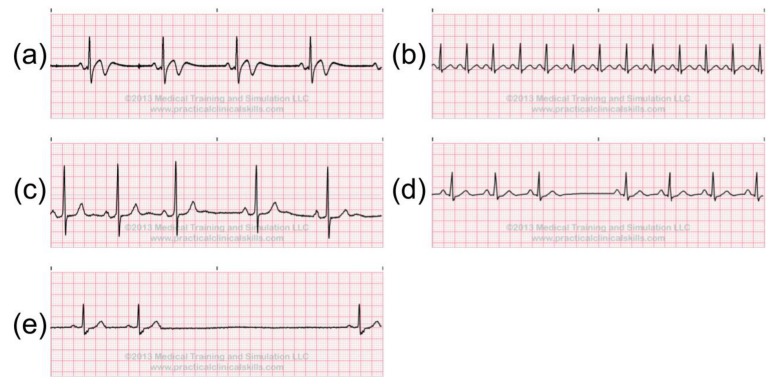
Abnormal sinus rhythms: (**a**) sinus bradycardia, (**b**) sinus tachycardia, (**c**) sinus arrhythmia, (**d**) sinus block, (**e**) sinus arrest [21].

**Figure 4 sensors-20-01461-f004:**
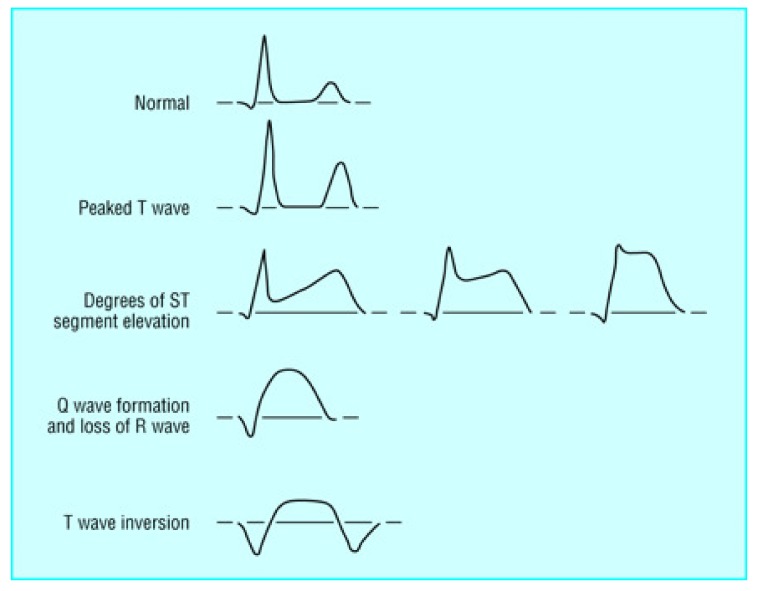
Examples of ST-T changes.

**Figure 5 sensors-20-01461-f005:**
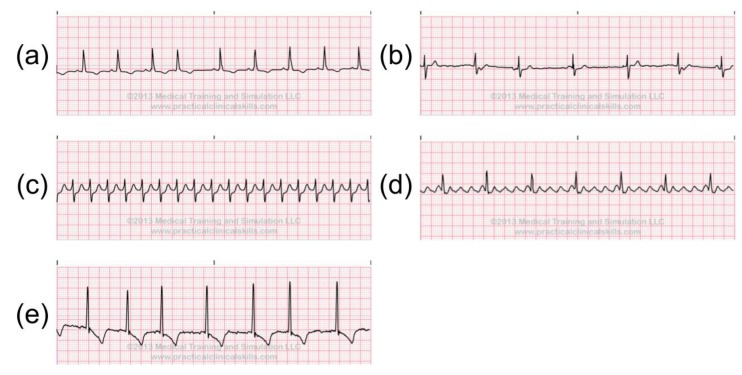
Abnormal Atrial Rhythms: (**a**) Premature Atrial Contraction, (**b**) Wandering Atrial Pacemaker, (**c**) Atrial Tachycardia, (**d**) Atrial Flutter, (**e**) Atrial Fibrillation [21].

**Figure 6 sensors-20-01461-f006:**
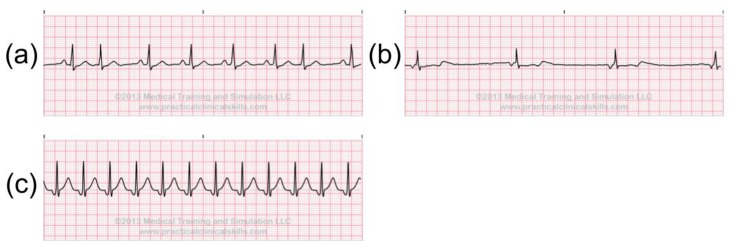
Abnormal Junctional Rhythms: (**a**) Premature Junctional Contraction, (**b**) Junctional Escaped Rhythm, (**c**) Junctional Tachycardia [21].

**Figure 7 sensors-20-01461-f007:**
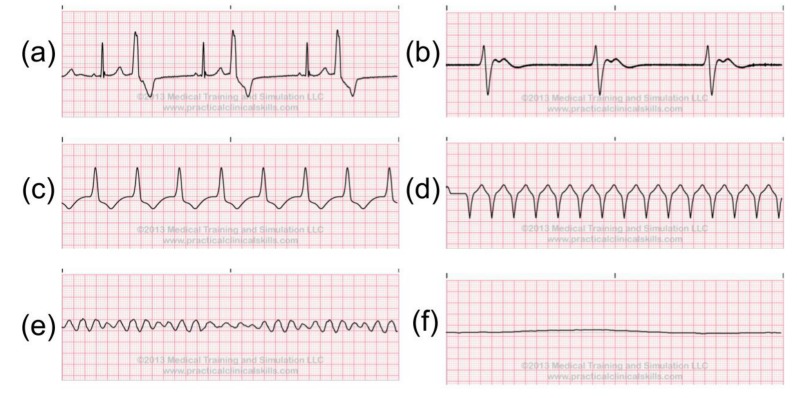
Abnormal Ventricular Rhythms: (**a**) Premature Ventricular Contraction, (**b**) Ventricular Escaped Rhythm, (**c**) Accelerated Idioventricular Rhythm, (**d**) Ventricular Tachycardia, (**e**) Ventricular Fibrillation, (**f**) Ventricular Asystole [21].

**Figure 8 sensors-20-01461-f008:**
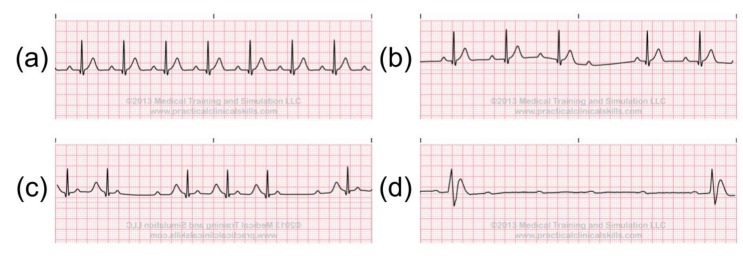
AV Blocks: (**a**) First-degree AV blocks, (**b**) Second-degree AV blocks type I, (**c**) Second-degree AV blocks type II, (**d**) Third-degree AV blocks [21].

**Figure 9 sensors-20-01461-f009:**
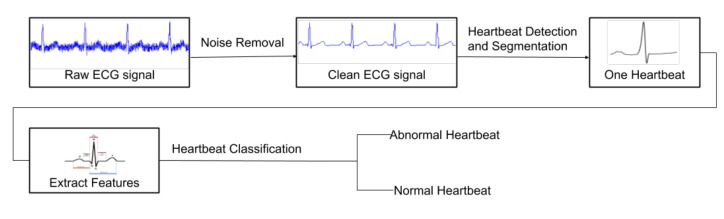
Typical Heartbeat Anomaly Detection.

**Figure 10 sensors-20-01461-f010:**
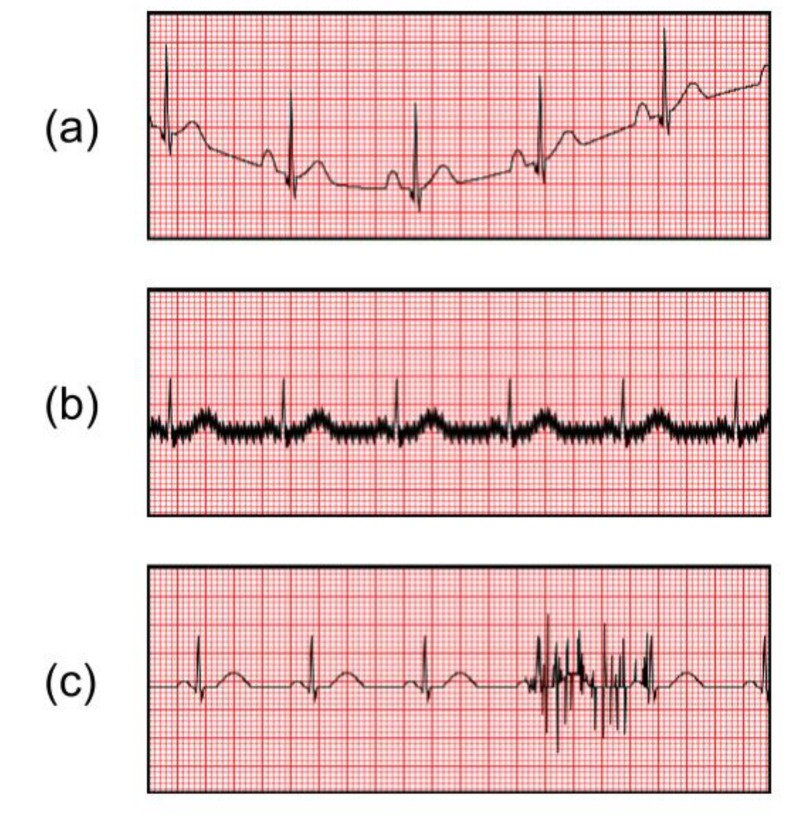
ECG Artifact examples: (**a**) Baseline Wander, (**b**) Power line Interference, (**c**) Muscle Interference.

**Figure 11 sensors-20-01461-f011:**
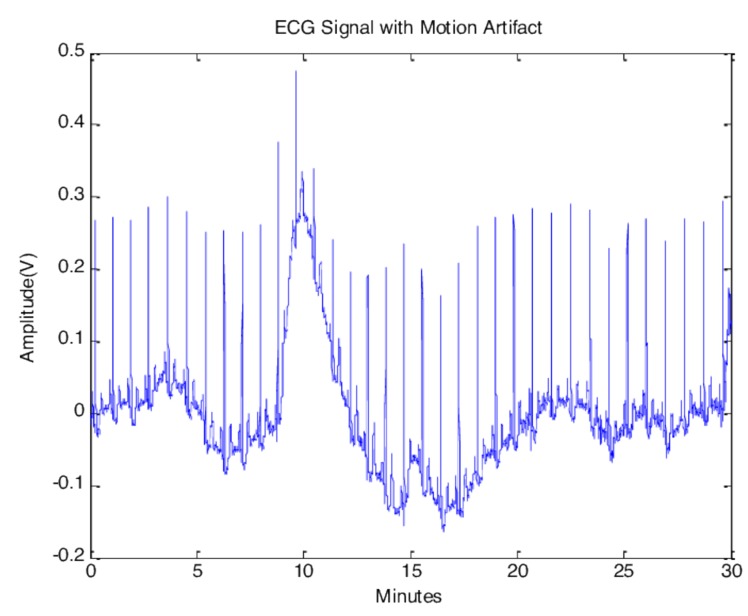
ECG Motion Artifact.

**Figure 12 sensors-20-01461-f012:**
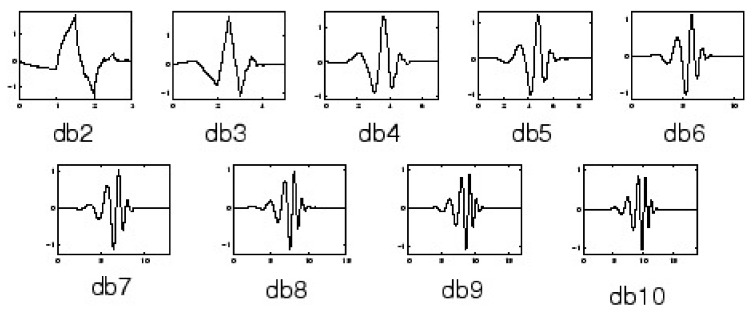
Daubechies wavelets.

**Figure 13 sensors-20-01461-f013:**
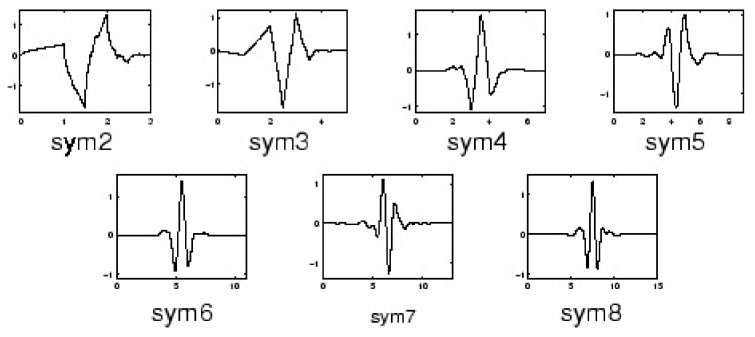
Symlet wavelets.

**Figure 14 sensors-20-01461-f014:**
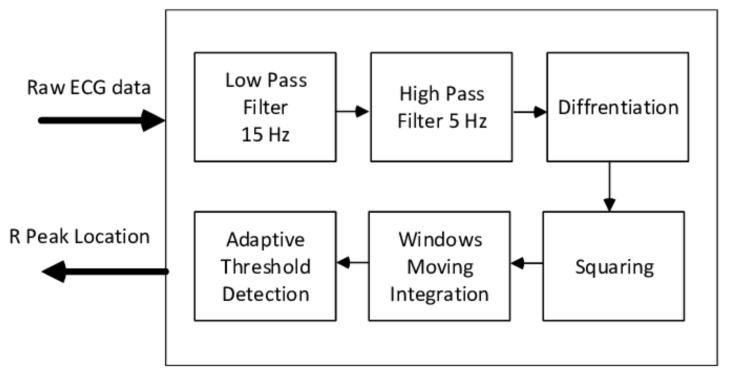
The Pan–Tompkins Algorithm.

**Figure 15 sensors-20-01461-f015:**
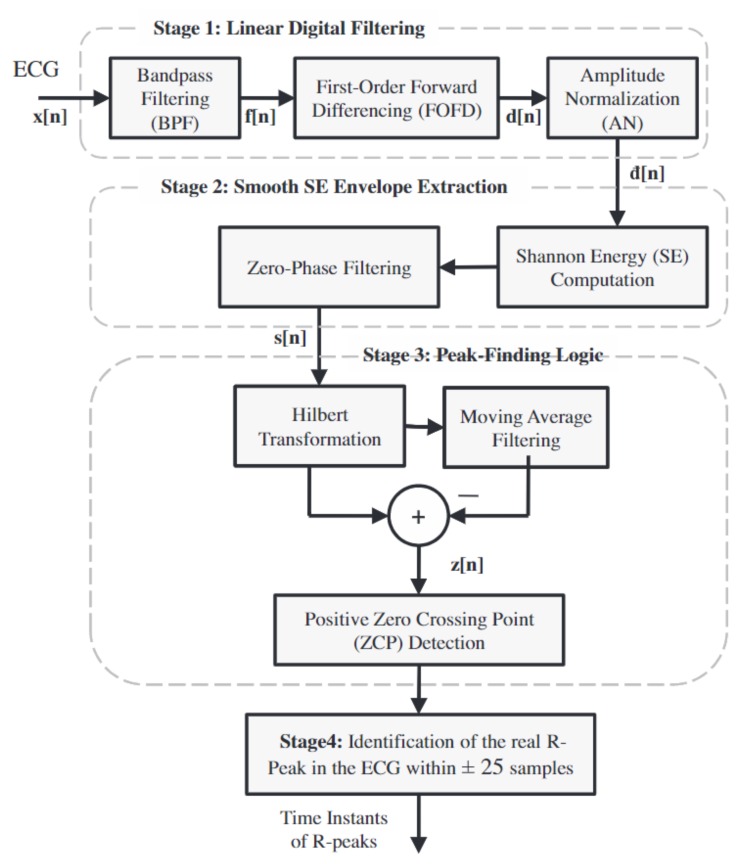
SEEHT R-peak detection algorithm.

**Figure 16 sensors-20-01461-f016:**
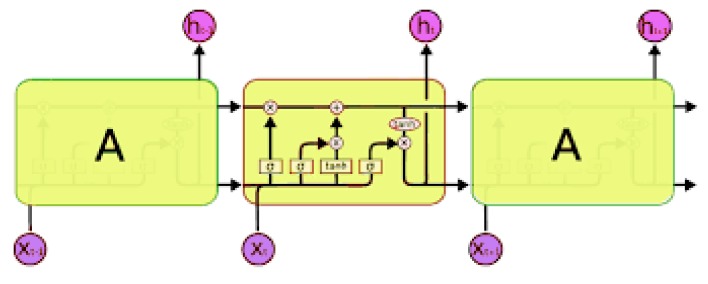
Long short-term memory layers.

**Table 1 sensors-20-01461-t001:** The heartbeat detection performance comparison using the MIT-BIH data set.

Method	Year	Total Heartbeats	TP	FP	FN	SEN	+P	DER	ACC
Pan–Tompkins [56]	1985	116,137	115,860	507	277	99.76%	99.56%	0.68%	99.33%
FBBBD [57]	1999	91,283	90,909	406	374	99.59%	99.56%	0.86%	99.15%
S.W.Chen [58]	2006	102,654	102,195	529	459	99.55%	99.49%	0.97%	99.04%
DOM [60]	2008	116,137	115,971	58	166	99.86%	**99.95**%	0.19%	99.81%
S.Choi [59]	2010	109,494	109,118	218	376	99.66%	99.80%	0.54%	99.46%
Z.Zidelmal [61]	2012	109,494	109,101	193	393	99.64%	99.82%	0.54%	99.47%
SEEHT [63]	2012	109,496	109,417	140	79	99.93%	99.87%	0.2%	99.80%
S.Banerjee [68]	2012	19140	19126	20	20	99.90%	99.90%	0.21%	99.79%
PSEE [64]	2013	109,494	109,401	91	93	99.92%	99.92%	0.17%	99.83%
F.Bouaziz [62]	2014	109,494	109,354	232	140	99.87%	99.79%	0.34%	99.66%
A.Karimipour [69]	2014	116,137	115,945	308	192	99.83%	99.74%	0.43%	99.57%
ISEE [65]	2016	109,532	109,474	116	58	**99.95**%	99.89%	**0.16**%	**99.84**%
WTSEE [66]	2017	109,494	109,415	99	79	99.93%	99.91%	**0.16**%	**99.84**%

**Table 2 sensors-20-01461-t002:** Conventional Morphological Features of Heartbeats.

Features	Description	Reference
QRS complex duration	The time interval between the onsite of the Q wave and offsite of the S wave	[30,31,38] [73,74,75]
QRS velociy left	The QRS slope velocity calculated for the time-interval between the QRS complex onset and the first peak	[30,73]
QRS velociy right	The QRS slope velocity calculated for the time-interval between the first peak and the second peaks	[30,73]
QRS complex area	The sum of the positive area and absolute negative area in the QRS complex	[30,73]
QRS complex morphology	Sample points from the QRS onsite to the QRS offsite	[31]
QRS complex AC power	The total power content of the QRS complex signal	[32]
QRS complex Kurtosis	The kurtosis indicates the peakedness of the QRS complex	[32]
QRS complex Skewness	The skewness measures the symmetry of the distribution of the QRS complex	[32]
Q wave valley	The valley value of Q wave	[75]
S wave valley	The valley value of S wave	[75]
T wave peak	The peak value of T wave	[75]
T wave duration	The duration from the QRS offsite to the T wave offsite	[31]
T wave morphology	Sample points from the QRS offsite to the T wave offsite	[31]
P wave flag	A Boolean value indicates the presence or absence of the P wave	[31]
P wave duration	The duration from the P wave onsite to the P wave offsite	[74]
P wave morphology	Sample points from the P wave onsite to the P wave offsite	[34,74]
PR interval duration	The duration from the P wave onsite to the QRS complex onsite	[74]
PR interval morphology	Sample points from the P wave onsite to the QRS complex onsite	[34]
QT interval duration	The duration from the QRS complex onsite to the T wave offsite	[74]
QT interval morphology	Sample points from the QRS complex onsite to the T wave offsite	[34,75]
ST interval morphology	Sample points from the S wave valley to the T wave offsite	[75]
Max peak(R peak) value	The maximum amplitude of the heartbeat	[30,73,75]
Min peak value	The minimum amplitude of the heartbeat	[30,73]
Positive QRS complex area	The area of the positive sample points in the QRS complex	[30,73,74]
Negative QRS complex area	The area of the negative sample points in the QRS complex	[30,73,74]
Positive P wave area	The area of the positive sample points in the P wave	[74]
Negative P wave area	The area of the negative sample points in the P wave	[74]
Positive T wave area	The area of the positive sample points in the T wave	[74]
Negative T wave area	The area of the negative sample points in the T wave	[74]
Absolute velocity sum	Sum of the absolute velocities in the pattern interval	[30,73]
Ima	Time-interval from the QRS complex onset to the maximal peak	[30,73]
Imi	Time-interval from the QRS complex onset to the minimal peak	[30,73]
Pre-RR interval	The RR interval between the heartbeat and its previous heartbeat	[31,71,74]
Post-RR interval	The RR interval between the heartbeat and its following heartbeat	[31,71,74]
Post-PP interval	The PP interval between the heartbeat and its following heartbeat	[74]
Average-RR interval	The average value of all valid RR intervals in the ECG record	[31,71,74] [32,75]
Local Average-RR interval	The average value of ten valid RR intervals surrounding the heartbeat	[31,71,74]
Normalized signal	The heartbeat sample points are normalized and down-sampled to have a mean of zero and standard deviation of one	[76,77,78]
Raw/downsampled ECG signal	The unprocessed ECG signal or the only processing on the signal is downsampled	[36,79]

**Table 3 sensors-20-01461-t003:** Conventional Derived Features of the Heartbeats.

Features	Method	Description	Reference
VCG amplitude	VCG	Maximal amplitude of the VCG vector	[30,38]
VCG sine angle	VCG	Sine component of the angle of the maximal amplitude vector	[30,38]
VCG cosine angle	VCG	Cosine component of the angle of the maximal amplitude vector	[30,38]
DTW distance	DTW	The Dynamic Time Warping distance between a heartbeat segment and the median heartbeat segment of the recording	[74,76]
Positive peak of the QRS complex	DWT	The positive peak amplitude of QRS complex on the fourth scale of the DWT	[38]
Negative peak of the QRS complex	DWT	The absolute negative peak amplitude of QRS complex on the fourth scale of the DWT	[38]
Positive peak of T wave	DWT	The positive peak amplitude of the T wave on the fourth scale of the DWT	[38]
Absolute T wave offsite	DWT	The absolute amplitude of the T wave offsite on the fourth scale of the DWT	[38]
R-S interval distance	DWT	The relative distance between the R peak and S valley on the fourth scale of the DWT	[38]
S-T interval distance 1	DWT	The relative distance between the S valley to the T wave peak on the fourth scale of the DWT	[38]
S-T interval distance 2	DWT	The relative distance between the S valley to the T wave offsite on the fourth scale of the DWT	[38]
Absolute maximum	DWT	The absolute maximum value and location on the fourth scale of the DWT signal	[38]
Zero crossing	DWT	The zero crossing location on the fourth scale of DWT signal	[38]
Wavelet scale	DWT	Calculate which scale the QRS complex is centered on	[38]
DWT coefficients	DWT	The down-sampled third and fourth detail coefficients and the fourth approximation coefficients	[71]
Independent Components	ICA	Independent components calculated with a fast fixed point algorithm	[71]
Fourier spectrum	DTCWT	Compute the absolute value of fourth and 5th scale DTCWT detail coefficients(dc). Then 1D FFT is applied to the selected DC to obtain the Fourier spectrum. Then take logarithm value of the Fourier spectrum	[32]
IMF sample entropy	EMD/EEMD	The sample entropy is measured of regularity of a time series used to quantify the complexity of heartbeat dynamics	[33]
IMF variation coefficient	EMD/EEMD	The coefficient of variation is a statistical parameter defined as $\sigma^{2}$/ $\mu^{2}$. ${}^{1}$	[33]
IMF singular values	EMD/EEMD	The singular value decomposition	[33]
IMF band power values	EMD/EEMD	The band power is the average power of each IMF	[33]
PCA components	PCA	PCA components for size reduction	[82]
Pisarenko PSD	Eigenvector	Power spectral density estimates generated with Pisarenko method	[83]
MUSCI PSD	Eigenvector	Power spectral density estimates generated with Multiple signal classification method	[83]
Minimum-Norm PSD	Eigenvector	Power spectral density estimates generated with Minimum-Norm methods	[83]

^1^σ is the standard variation of the selected IMF, m is the mean of the selected IMF.

**Table 4 sensors-20-01461-t004:** Heartbeat classification performance on the MIT-BIH dataset.

Method	Year	Abnormal/Normal	Heartbeat Types	TP	FP	TN	FN	Sensitivity	False Alarm	Accuracy
Christov et al. [30]-morphology	2006	18,378/47,239	5	180,42	1604	45,635	336	98.17%	3.40%	97.04%
Christov et al. [30]-frequency	2006	18,378/47,239	5	17,590	1459	45,780	788	95.71%	3.09%	96.58%
Chazal et al. [31]-frequency	2006	4317/34,394	5	4108	1962	32,432	209	95.16%	5.70%	94.39%
Ubeyli et al. [83]	2009	269/90	4	268	2	88	2	99.26%	2.22%	99.89%
Llamedo et al. [38]	2010	5441/44,188	3	4752	2238	41,950	689	87.34%	5.06%	94.10%
Ye et al. [71]-rejection	2012	19,913/64,042	16	19,815	93	63,949	98	99.51%	0.15%	99.77%
Ye et al. [71]-bayesian	2012	20,745/65,264	16	20,557	286	64,978	188	99.09%	0.44%	99.45%
Zhang et al. [74]	2014	5653/44,011	4	5248	4869	39,142	405	92.84%	11.06%	89.38%
Thomas et al. [32]	2015	26,626/672,68	5	22,900	1300	65,968	3726	86.01%	1.93%	94.65%
Kiranyaz et al. [36]	2015	7366/42,191	5	6539	1228	40,963	827	88.77%	2.97%	95.85%
Rajesh et al. [33]	2017	8000/2000	5	7677	33	1967	323	95.96%	1.65%	96.44%
Sahoo et al. [75]	2017	807/244	4	798	5	239	9	98.88%	2.04%	98.67%

**Table 5 sensors-20-01461-t005:** Rhythm classification performance on the MIT-BIH dataset.

Method	Year	Abnormal/Normal	Rhythm Types	Rhythm Length	TP	FP	TN	FN	Sensitivity	False Alarm	Accuracy
Ge et al. [94]	2002	713/143	6	1.2 s	706	10	133	7	88.77%	6.99%	98.01%
U. Acharya Net A [40]	2017	20807/902	4	2 s	19,160	62	840	1647	92.08%	6.87%	92.13%
U. Acharya Net B [40]	2017	8322/361	4	5 s	7946	376	294	67	95.48%	18.56%	94.9%

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
