# Peer review of "A Survey of Heart Anomaly Detection Using Ambulatory Electrocardiogram (ECG)"

_sensors, 2020, doi:10.3390/s20051461_

Round 1
Reviewer 1 Report
This paper reviewed the current state-of-the-art technology about heart anomaly detection based on ambulatory electrocardiogram (ECG). In our opinion, the logical organization of this paper, as well as the English language could be improved. The major concerns are as follows:
1. In abstract, the current situation of cardiovascular diseases should be updated using the latest published data.
2. The authors said “This paper review the current state-of-the-art of this technology … and the possible research directions to develop the next generation of ambulatory monitoring systems.” However, there was no next generation research directions introduced in this article.
3. In the Introduction L21, the most recent wearable single-lead ECG monitoring devices should be reviewed.
4. In page 2, L62-64, how [11]-[13] show it is possible to reconstruct standard 12-lead ECG? We suggest to use theoretical considerations of heart dipole models in the justification of this issue.
5. Section 2.2 should be further revised. We suggest authors should review not only the neural network based ECG synthesis methods, but also the linear model based synthesis methods, since the linear regression has been the most often utilized method, the results of linear method is competitive. The following articles could help:
[1] Frank E. An Accurate, Clinically Practical System For Spatial Vectorcardiography. Circulation, 1956, 13(5):737.
[2] Uijen GJH., Van Oosterom A., Van Dam R. The relationship between the 12-lead standard ECG and the XYZ vector leads. in Proc. 14th Int. Cong. Electrocardiol. Berlin. 1988, pp. 301–307.
[3] Nelwan SP., Kors JA., Meij SH., Van Bemmel JH., Simoons ML. Reconstruction of the 12-lead electrocardiogram from reduced lead sets. J. Electrocardiol. 2004, 37(1):11–18.
[4] Figueiredo, C.P., Mendes, P.M.: Towards wearable and continuous 12-lead electrocardiogram monitoring: Synthesis of the 12-lead electrocardiogram using 3 wireless single-lead sensors. In: Proceedings of the International Conference on Biomedical Electronics and Devices (BIODEVICES), pp. 329–332 (2012)
[5] Tomasic, I., Trobec, R., Lindén, M.: Can the regression trees be used to model relation between ECG leads? In: Internet of Things. IoT Infrastructures: Second International Summit, pp. 467–472. Springer International Publishing (2016)
[6] Zhu H., Pan Y., Cheng K.T., et al. A lightweight piecewise linear synthesis method for standard 12-lead ECG signals based on adaptive region segmentation. PLOS ONE, 2018, 13(10): e0206170.
6. What is the relationship between irregular heart rate, irregular heart rhythm, ectopic rhythms, Junctional Rhythms, and Ventricular Rhythms in Sec. 2.3. Further explanations are required.
7. In Sec. 3.1, the authors said “Therefore, an anomaly detection system is composed of four different sub-systems: heartbeat detection, heartbeat segmentation, heartbeat classification, and rhythm classification (see in Fig.8)”. However, those sub-systems could not be seen in Fig. 8. Moreover, the authors did not organize article by those sub-systems in the following sections.
8. L227-229, is the character of IMF related to the noise removal of ECG? Please justify.
9. It seems that Table 1 only concluded the heart beat detection. Where is the conclusion of heartbeat segmentation in L284-297?
10. What is the difference between heart beat detection and beat detection? So as heartbeat classification and beats classification. Please unify all the professional terms in this paper.
11. Is “the computed features” referred to morphological features, derived features, or both?
12. Table 2 and Table 4 were beyond the page. And Table 4 appeared after Table 5.
13. Some numbers are missing, i.e. L361 “Minimum-Norm method [? ]”, L566 “heartbeat segmentation (Section ??)”, and the reference number of “Thakor et. al” in Page 10.
14. Eq.(4) seems to be wrong. Please justify.
15. What were the accuracies of those work introduced in L415-425?
16. L8 “review”=> “reviews” or “reviewed” ;L30, “In odder to”=>”In order to”; L494, “N x P” => “N * P”; L 499, “to built”=> “to build”; L504 “transform” => “transformation”.
Author Response
Dear Reviewer,
Thanks for your valuable suggestion. I have modified my paper accordingly. The revised the paper with highlighting the revised part is in the attachment. I have written my response below as well.
- In abstract, the current situation of cardiovascular diseases should be updated using the latest published data.
- the current situation of CVD has been updated according to the world health organization in both abstract and introduction.
- The authors said “This paper review the current state-of-the-art of this technology … and the possible research directions to develop the next generation of ambulatory monitoring systems.” However, there was no next generation research directions introduced in this article.
- An new text was added to describe the future trends in patient monitoring using sensor fusion and redundant neural networks.
- In the Introduction L21, the most recent wearable single-lead ECG monitoring devices should be reviewed.
- Two 1-lead ECG monitoring devices has been added. However, the majority research has been done on the MIT-BIH database, and I did not find much research that is doing anomaly detection using the mention wearable ECG monitoring device.
- In page 2, L62-64, how [11]-[13] show it is possible to reconstruct standard 12-lead ECG? We suggest to use theoretical considerations of heart dipole models in the justification of this issue.
- This part was not the main focus of my review. It intended to show that many publications show that 3-lead ECG contain enough information for anomaly detection.
- Section 2.2 should be further revised. We suggest authors should review not only the neural network based ECG synthesis methods, but also the linear model based synthesis methods, since the linear regression has been the most often utilized method, the results of linear method is competitive.
- Added a literature that uses linear model to reconstruct the 12 lead ECG signal. I did not review more literature in this section because this part was not the main focus of my review. It intended to show that 3-lead ECG contain enough information for anomaly detection. So I was trying to make it more succinct.
- What is the relationship between irregular heart rate, irregular heart rhythm, ectopic rhythms, Junctional Rhythms, and Ventricular Rhythms in Sec. 2.3. Further explanations are required.
- The hear rated o not necessarily has a relationship with the rhythms. The ectopic rhythms contain atrial rhythms, junctional rhythms, and ventricular rhythm. The relationship of the rhythms was mentioned in the paper.
- In Sec. 3.1, the authors said “Therefore, an anomaly detection system is composed of four different sub-systems: heartbeat detection, heartbeat segmentation, heartbeat classification, and rhythm classification (see in Fig.8)”. However, those sub-systems could not be seen in Fig. 8. Moreover, the authors did not organize article by those sub-systems in the following sections.
- Added noise removal to the sub-systems. And also explained why there is a section for MIT-BIH database.
- L227-229, is the character of IMF related to the noise removal of ECG? Please justify.
- Added explanation of how does EMD remove noise, and added signal reconstruction equation.
- It seems that Table 1 only concluded the heart beat detection. Where is the conclusion of heartbeat segmentation in L284-297?
- There wasn't a table for heartbeat segmentation algorithms comparison, since there isn't a valid metric to compare them. However, heartbeat segmentation literature is introduced in the end of Section 3.3
- What is the difference between heart beat detection and beat detection? So as heartbeat classification and beats classification. Please unify all the professional terms in this paper.
- They are the same. The terms are unified.
- Is “the computed features” referred to morphological features, derived features, or both?
- It means both. I have changed the terms.
- Table 2 and Table 4 were beyond the page. And Table 4 appeared after Table 5.
- The size of the tables are reduced. The table 4 is before table 5 in my file. The table 4 is the heartbeat classification, and the table 5 is the rhythm classification.
- Some numbers are missing, i.e. L361 “Minimum-Norm method [? ]”, L566 “heartbeat segmentation (Section ??)”, and the reference number of “Thakor et. al” in Page 10.
- They are fixed
- Eq.(4) seems to be wrong. Please justify.
- Equation is fixed
- What were the accuracies of those work introduced in L415-425?
- The accuracy have been added. And some of them are compared in the table.
- L8 “review”=> “reviews” or “reviewed” ;L30, “In odder to”=>”In order to”; L494, “N x P” => “N * P”; L 499, “to built”=> “to build”; L504 “transform” => “transformation”.
- They are fixed.

Reviewer 2 Report
The authors have presented a review for the Heart Anomaly Detection Using
Ambulatory Electrocardiogram (ECG). As a review paper it brings value to readers, especially young researchers and PhD students interested in this research topic.
The authors should focus a little bit more on:
(1) real time aspect, and how the classified event can be triggered to professionals
(2) the data rate should be also discussed, especially if it has to be complemented with other data streams for vital parameters
(3) The title should read better 'Review' or 'Survey' instead of 'Overview.
Author Response
Dear Reviewer,
Thanks for your valuable suggestion. I have modified my paper accordingly. The revised the paper with highlighting the revised part is in the attachment. I have written my response below as well.
-
real time aspect, and how the classified event can be triggered to professionals
- The mentioned wearable ECG device requires manually upload the ECG signal to its cloud server that could do data analysis and management. The monitoring time could be from 1 min to 24 hours for those devices.
-
the data rate should be also discussed, especially if it has to be complemented with other data streams for vital parameters
- As mentioned in the 1st question, the user manually upload their ECG signal to the server. The data rate is depends on how often they upload their data.
-
The title should read better 'Review' or 'Survey' instead of 'Overview.
- The tile has been changed

Reviewer 3 Report
The authors have done a lot in literature review. The authors not only introduced the detection system and algorithm, but also explained how to judge various abnormal signals in medicine. But I think the article still needs some modification.
(line 55) What is the PEM? A scheme of the relationship between heart structure and ECG signals should be suppled. In 3.1 (line 136), the authors mentioned that an anomaly detection system contains four sub-systems. But many methods using deep learning, including the literature cited by authors, do not have all four systems. The authors should have a better description. Can figure 11 and 12 be replaced with denoised signals? Please resize tables 2 and 4. In 3.4.2 (line 434), please specify which 4 heartrate types. In 3.5 (line 499 and 510), please specify the classification accuracy. In 4.1 (line 546), the authors mentioned that moving will introduce noise, but this noise is easily identified by human. However, to my knowledge, much of the research work on heart anomaly detection is based on existing dataset. In other words, the data is measured without continuous movement. Moreover, in the article, the authors did not mention the research on the heart anomaly detection during moving. Therefore, I think the first challenge is unreasonable. In 4.1 (line 554), the authors mentioned an imbalanced dataset is a challenge. The authors should provide the solution to the problem. Please cite some literature or explain how published papers solve this problem. In reference (line 595, 596, 597), the website of the devices should be given. In the 3, 4, 5 sections, there are less figures. We think authors should choose some representative models and review it with figures. It is beneficial to the new researchers of this field.
Author Response
Dear Reviewer,
Thanks for your valuable suggestion. I have modified my paper accordingly. The revised the paper with highlighting the revised part is in the attachment. I have written my response below as well.
- (line 55) What is the PEM? A scheme of the relationship between heart structure and ECG signals should be suppled.
- PEM is portable ECG monitor. The changes has been made in the paper.
- In 3.1 (line 136), the authors mentioned that an anomaly detection system contains four sub-systems. But many methods using deep learning, including the literature cited by authors, do not have all four systems. The authors should have a better description.
- Added noise removal to the sub-systems. And also explained why there is a section for MIT-BIH database.
- Can figure 11 and 12 be replaced with denoised signals? Please resize tables 2 and 4.
- The figures were intend to show the most popular mother wavelet basis function for discrete wavelet transform that is used for filtering the ECG signal.
- In 3.4.2 (line 434), please specify which 4 heartrate types.
- The four heartbeat types are specified.
- In 3.5 (line 499 and 510), please specify the classification accuracy.
- The accuracy have been added. And some of them are compared in the table.
- In 4.1 (line 546), the authors mentioned that moving will introduce noise, but this noise is easily identified by human. However, to my knowledge, much of the research work on heart anomaly detection is based on existing dataset. In other words, the data is measured without continuous movement. Moreover, in the article, the authors did not mention the research on the heart anomaly detection during moving. Therefore, I think the first challenge is unreasonable.
- Most research discussed in the paper tested their methods on the MIT-BIH database which is collected using Holter device. The other mentioned wearable ECG devices are also Holter devices. Even though the MIT-BIH databse is not affected by motion artifacts, but if people are using the wearable ECG device during their daily activities, the collected signal will be contaminated with motion artifact. In addition, there is not much research that has analyzed the database with motion artifact. And the database with motion artifact is lacking as well.
- In 4.1 (line 554), the authors mentioned an imbalanced dataset is a challenge. The authors should provide the solution to the problem. Please cite some literature or explain how published papers solve this problem.
- I have added several possible solution for imbalanced dataset. However, this part was intended to be a direction for the future research.
- In reference (line 595, 596, 597), the website of the devices should be given.
- reference are fixed.
- In the 3, 4, 5 sections, there are less figures. We think authors should choose some representative models and review it with figures. It is beneficial to the new researchers of this field.
- added figures for heartbeat detection algorithm. And also added some figures for some deep learning model.

Reviewer 4 Report
Dear Authors, I would first congratulate you for your extensive work. Yet, your manuscript needs some improvements so that it becomes suitable for publication in Sensors: 1. Line 2: More recent data about CV mortality (2018/2019?). The same in line 13 of the manuscript 2. Atrial fibrillation is usually abbreviated in medical papers as “AF”. 3. Line 55: Please, spell “PEM” the first time it appears in the article 4. The entire 2.4 section “Abnormal ECG Signals” is superfluous. 5. Generally, the entire manuscript is overextended to be a journal article. Please, make it more succinct. You can omit 80-90% of the information in sections 1 and 2, and you should select/extract the most important data from section 3 so that your article be accommodated into 6-8 pages (10 pages at most!). Please, keep in mind that it is not a chapter of a textbook or a PhD work.Author Response
Dear Reviewer,
Thanks for your valuable suggestion. I have modified my paper accordingly. The revised the paper with highlighting the revised part is in the attachment. I have written my response below as well.
- Line 2: More recent data about CV mortality (2018/2019?).
- the current situation of CVD has been updated according to the world health organization in both abstract and introduction.
- Atrial fibrillation is usually abbreviated in medical papers as “AF”.
- The abbreviated term has been fixed.
- Line 55: Please, spell “PEM” the first time it appears in the article
- "PEM" is portable ECG monitor. Changes are also been addressed in the article.
- The entire 2.4 section “Abnormal ECG Signals” is superfluous.
- Our goal is to present the types of anomaly one needs to detect for non-medical readers
- Generally, the entire manuscript is overextended to be a journal article. Please, make it more succinct. You can omit 80-90% of the information in sections 1 and 2, and you should select/extract the most important data from section 3 so that your article be accommodated into 6-8 pages (10 pages at most!). Please, keep in mind that it is not a chapter of a textbook or a PhD work.
- Our goal is to do a review for an non-medical audience and many of the lit presented is essential to the integrity of the paper.

Round 2
Reviewer 1 Report
We check the manuscript, and some problems still remain after the revision.
1. We agree “3-lead ECG contain enough information for anomaly detection”, and the 12-lead ECG synthesis was not the main focus of the review. However, the reason why it is possible to reconstruct standard 12-lead ECG, as well as the most recent synthesis methods should be reviewed as the technical background of ECG anomaly detection using 3-lead ECG. Even for simplicity, the only literatures chosen about this issue, e.g., [12], [15], [16], in the revision is not adequate or state-of-the-art.
2. In Fig.8, "subsystem" and "rhythm classification" are still not included.
3. Eq.5 (as the Eq.4 in last version) seems still wrong, if w(n) is a N by 1 vector.
Author Response
Dear Reviewer,
Thanks for your valuable suggestion. I have modified my paper accordingly. The revised the paper with highlighting the revised part is in the attachment. I have written my response below as well.
- We agree “3-lead ECG contain enough information for anomaly detection”, and the 12-lead ECG synthesis was not the main focus of the review. However, the reason why it is possible to reconstruct standard 12-lead ECG, as well as the most recent synthesis methods should be reviewed as the technical background of ECG anomaly detection using 3-lead ECG. Even for simplicity, the only literatures chosen about this issue, e.g., [12], [15], [16], in the revision is not adequate or state-of-the-art.
- Added more literature reviews for section 2.2
- In Fig.8, "subsystem" and "rhythm classification" are still not included.
- In Fig.8(now is Fig.9), the subsystems are noise removal, heartbeat detection and segmentation, and heartbeat classification. This figure shows the steps for heartbeat anomaly detection. The rhythm classification is very similar to heartbeat classification, instead of dealing with one heartbeat, it deals with several heartbeats. Normally, the heartbeat classification and rhythm classification do not co-exist, because they are dealing with different kind of data.
- Eq.5 (as the Eq.4 in last version) seems still wrong, if w(n) is a N by 1 vector.
- eq5. is fixed. Forgot to add bar on top of w. Now the equation is identical to the equation in the reference.

Reviewer 4 Report
Dear Authors,
Your manuscript has been improved significantly. It could now be considered for publication. I suggest just 2 additional minor corrections of technical character:
In the abstract, the second sentence – “….died from CVD….” Should be “…die from CVD…”. The same in the “Introduction”.
In section ”2.4 Abnormal ECG Signals: There are two kinds of anomalies in ECG signals” - I suggest that you add “rhythm” to “anomalies” because changes in ST-segment and T-wave could also be anomalies associated with pathological conditions different from dysrhythmias.
Author Response
Dear Reviewer,
Thanks for your valuable suggestion. I have modified my paper accordingly. The revised the paper with highlighting the revised part is in the attachment. I have written my response below as well.
- In the abstract, the second sentence – “….died from CVD….” Should be “…die from CVD…”. The same in the “Introduction”.
- The grammar is fixed.
- In section ”2.4 Abnormal ECG Signals: There are two kinds of anomalies in ECG signals” - I suggest that you add “rhythm” to “anomalies” because changes in ST-segment and T-wave could also be anomalies associated with pathological conditions different from dysrhythmias.
- The anomalies have three subsets now: irregular heart rate, irregular rhythm, and ecotopic rhythm. A new paragraph and a new figure have added for irregular rhythm as well.
